# Sparse Subspace Diffusion Model for Physically Consistent Accelerated MRI Reconstruction

**Xiangyao Deng**[1]                                                    XIANGYAO.DENG@MONASH.EDU
**Zhiqiang Shen**[1]                                                         XXSZQYY@GMAIL.COM
**Sanuwani Dayarathna**[1]                              SANUWANI.HEWAMUNASINGHE@MONASH.EDU
**Juan P. Meneses**[2]                              JUANPABLO.MENESESCASANOVA@MONASH.EDU
**Sergio Uribe**[2]                                                    SERGIO.URIBE@MONASH.EDU
**Zhaolin Chen**[1,3]                                                  ZHAOLIN.CHEN@MONASH.EDU

[1] *Department of Data Science and AI, Monash University, Melbourne, Australia*

[2] *Department of Medical Imaging and Radiation Sciences, Monash University, Melbourne, Australia*

[3] *Monash Biomedical Imaging, Monash University, Melbourne, Australia*

**Editors:** Under Review for MIDL 2026

## Abstract

Magnetic resonance imaging (MRI) provides excellent soft-tissue contrast but suffers from long acquisition times. Accelerated MRI alleviates this issue by undersampling k-space, but this approach introduces aliasing artifacts and information loss. Traditional compressed sensing methods exploit handcrafted sparse priors, whereas deep learning approaches learn data-driven priors, but both often struggle at high acceleration rates due to severe information degradation. This study introduces a diffusion-based reconstruction framework, termed the Sparse Subspace Diffusion Model (SSDM), that performs MRI reconstruction within an adaptive sparse space. The proposed approach integrates coupling convolutional dictionary learning with diffusion-based generative modeling to decompose MR images into multiple orthogonal sparse subspaces and reconstruct them under measurement-consistency constraints. This formulation enables diffusion modeling in a physically meaningful latent space, effectively bridging the gap between data-driven learning and physics-guided reconstruction. Experimental results on the fastMRI dataset demonstrate that the proposed method achieves higher reconstruction quality than existing diffusion- and sparsity-based approaches, with better preservation of fine details and suppression of artifacts across various acceleration factors.

**Keywords:** Magnetic Resonance Imaging (MRI), fastMRI, Diffusion model, Convolutional Dictionary Learning, Physics-informed learning.

## 1. Introduction

Magnetic resonance imaging (MRI) provides excellent soft-tissue contrast but is limited by lengthy acquisition times. Accelerating the imaging process through undersampling the signal would cause aliasing artifacts and information loss. Compressed sensing (CS) (Lustig et al., 2007) constrains image reconstruction with sparse priors to recover high-quality images from incomplete data, but its reliance on handcrafted transforms limits adaptability across different scanning scenarios. In recent years, deep learning-based MRI reconstruction methods have significantly improved reconstruction performance by learning data-driven

priors (Schlemper et al., 2017). Generative models have been applied to MRI reconstruction, achieving high-fidelity inversion by learning image priors and incorporating measurement consistency constraints (Chung and Ye, 2022; Chung et al., 2024, 2022; Wang et al., 2022). Among these, measurement-consistent unconditional diffusion models require no paired training data and can generalize across multiple sampling patterns, thus demonstrating significant advantages in accelerated MRI.

In accelerated magnetic resonance imaging (MRI), aliasing artifacts arise from severe signal mixing caused by undersampling, posing significant challenges for both discriminative and generative reconstruction models. Under such conditions, energy leakage from the true signal leads to strong entanglement between aliasing components and real anatomical structures, rendering reliable signal recovery highly ill-posed. Meanwhile, MR images lie in a complex representation space comprising multiple features or frequency components. Conventional deep learning methods (Esser et al., 2021) typically rely on convolutional operations to map images into highly entangled latent representations, where different feature components lack sufficient separability. As a result, these models often produce perceptually plausible but feature-inconsistent reconstructions, without guaranteeing faithful recovery of the underlying signal components (Antun et al., 2020). Motivated by these observations, we adopt a divide-and-conquer strategy that explicitly decomposes the complex image representation into multiple sparse subspaces and performs reconstruction independently within each subspace. In prior work, Sub-DM(Guan et al., 2024) places different frequency components in the channel dimension via wavelet decomposition and jointly models them within a single diffusion trajectory, primarily to reshape the data geometry and stabilize the score field. Generative subspace diffusion(Jing et al., 2022) defines subspaces as noise-scale-dependent nested supports that progressively expand during sampling. In contrast, we redefine sparse subspaces as independent learning instances, reducing the modeling burden on any single instance. Specifically, each image is decomposed into multiple sparse subspaces, organized as independent samples along the batch dimension, modeled in parallel by a shared diffusion backbone, and their evolution is explicitly coupled through cross-batch attention. Specifically, we construct independent sparse subspaces via coupled convolutional dictionary learning (Yang et al., 2012), transforming the original high-complexity global modeling problem into a set of low-complexity subspace reconstruction tasks. To maintain global structural consistency, we further introduce a lightweight cross-subspace attention mechanism into our backbone to facilitate information exchange among subspaces during reconstruction.

Based on this design, we introduce a Sparse Subspace Diffusion Model (SSDM). This two-stage MRI reconstruction framework integrates coupled convolutional dictionary learning with diffusion-based generative modeling in an adaptive sparse space. Through a measurement-consistent guided reverse-diffusion process, the proposed framework achieves effective subspace-wise reconstruction and enables high-quality MRI reconstruction under severe undersampling. Experimental results on the fastMRI dataset demonstrate that, compared to existing diffusion-based reconstruction methods, the proposed framework achieves the advantages in both reconstruction fidelity and artifact suppression.

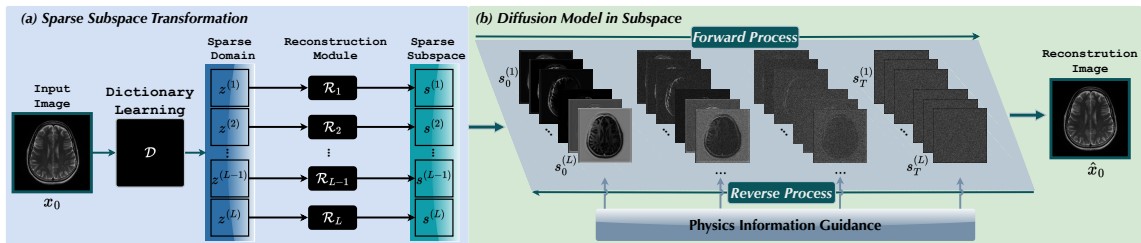

Figure 1: **Overview of the proposed sparse subspace diffusion model.** (a) The sparse transformation of the undersampled image provides physics-based guidance to the diffusion process through the measurement consistency constraint; (b) The unconditional diffusion model is trained on fully sampled subspaces in the forward process and, during the reverse process, generates subspaces guided by physics-informed priors from the undersampled image.

## 2. Methods

The forward model of MRI reconstruction is written as follows,

$$y = Ax + \epsilon. \tag{1}$$

The measurement matrix $A = M \odot \mathcal{F} \in \mathbb{C}^{m \times n}$ models undersampled acquisition, where $m \ll n$, $\mathcal{F}$ represents the discrete Fourier transform, and $M$ is a binary sampling mask in k-space. $x$ is the desired image and $\epsilon$ is the complex Gaussian noise. The $y$ is the measurement signal in k-space. To address the complexity of MRI image representation, we propose a sparse subspace diffusion model, which includes 1) coupled convolutional dictionary learning to decompose each image into multiple sparse and independent subspaces [Fig. 1(a)] and 2) diffusion modeling in subspace to train an unconditional diffusion model [Fig. 1(b)].

**Convolutional dictionary learning:** We employ convolutional dictionary learning (CDL) as an adaptive sparse transform to transfer MR images acquired at arbitrary acceleration factors into a unified multi-channel sparse domain. We unfold the dictionary learning process through the Learned Iterative Shrinkage-Thresholding Algorithm (LISTA) (Gregor and LeCun, 2010). To enhance the model's adaptability, we replace the original learnable soft-thresholding function with a spatial attention module that adaptively determines the threshold for each pixel based on the input signal. This design allows the dictionary learning model to adapt to image sparse decomposition under arbitrary acceleration factors. Each sparse channel can then be inversely transformed to the image space through the $i$-th single convolutional kernel in $\mathcal{R}$, forming a set of sparse subspaces that capture distinct structural features. We introduce the $i$-th subspace $s^{(i)} \in \mathbb{C}^{C \times H \times W}$ as,

$$s^{(i)} = \mathcal{R}^{(i)}\big(\mathcal{D}(x)\big), \forall i \in [1, L], \tag{2}$$

where $\mathcal{D}$ denotes the CDL-based adaptive sparse transform (Deng and Dragotti, 2020; Yang et al., 2025) that transforms an image into the sparse domain. $\mathcal{R}$ represents the reconstruction module that includes a transposed convolution operation to transform the

sparse representation back to the image domain, and $\mathcal{R}^{(i)}$ corresponds to the transposed convolutional operation with the $i$-th convolutional kernel.

To establish a coupling relationship between the undersampled and fully sampled subspaces in k-space based on the measurement-observed locations, we introduce a channel-wise coupling constraint. This coupling is governed by the sampling mask, which enforces alignment between the two subspaces at observed locations while actively suppressing the signal generation at unobserved positions. Furthermore, an orthogonality constraint is imposed among the sparse subspaces to enforce subspace decoupling and eliminate redundant information in the sparse representation. The overall CDL algorithm is given as,

$$\min_{\mathcal{R},\mathcal{D}} \parallel \mathcal{R}(\mathcal{D}(x)) - x \parallel_2^2 + \parallel \mathcal{D}(x) \parallel_1 + \mathcal{L}_\mathcal{C} + \mathcal{L}_\perp \tag{3}$$

The orthogonal constraint $\mathcal{L}_\perp$ minimizes the cosine angles amongst the subspaces $s$ in the vector space. The coupling constraint $\mathcal{L}_\mathcal{C}$ is written as,

$$\mathcal{L}_\mathcal{C} = \mathcal{L}_\Omega + \mathcal{L}_u, \tag{4}$$

which includes two terms. The first term, $\mathcal{L}_\Omega$, couples the signals in k-space based on the observation location along the channel dimension, across the under-sampled and full-sampled images. $\mathcal{L}_\Omega$ can be written as,

$$\mathcal{L}_\Omega = \left\| M \odot \left( \mathcal{F}(s_u) - \mathcal{F}(s_f) \right) \right\|_2^2 \tag{5}$$

where $s_u$ and $s_f$ denote the subspace of the undersampled image and the full-sampled image, respectively. To further constrain the representation in each sparse subspace, we introduce $\mathcal{L}_u$, which penalizes spurious energy in the unobserved k-space regions, given as,

$$\mathcal{L}_u = \parallel A(\sum_i^L s^{(i)}) - y \parallel_2^2 + \lambda_u \sum_i^L \left\| (1 - M) \odot \mathcal{F}(s^{(i)}) \right\|_2^2. \tag{6}$$

where $\lambda_u$ controls the strength of energy suppression in the unobserved k-space regions for each subspace. The first-stage CDL is fully trained before the second-stage diffusion modeling.

**Diffusion modeling in subspace:** At the second stage, we model the $i^{th}$ subspace $s^{(i)}$ using a diffusion model,

$$p_\theta(s^{(i)}) = \int p_\theta(s_T^{(i)}) \prod_{t=1}^T p_{\theta,t}\left( s_{t-1}^{(i)} \mid s_t^{(i)} \right) ds_{1:T}^{(i)}. \tag{7}$$

Unfolding the Markovian forward process as follows,

$$q(s_t^{(i)}|s_0^{(i)}) = \mathcal{N}(s_t^{(i)}; \sqrt{\bar{\alpha}_t}s_0^{(i)}, (1 - \bar{\alpha}_t)I), \ \bar{\alpha}_t = \prod_{\tau=1}^t \alpha_\tau \tag{8}$$

where the noise schedule $\alpha$ is a decreasing sequence of $t$. The diffusion model is trained by,

$$\min_\theta \sum_{i=1}^L \mathbb{E}_{\epsilon \sim \mathcal{N}(0,I)} \left[ \parallel \epsilon_\theta(s_t^{(i)}, t) - \epsilon^{(i)} \parallel_2^2 \right] + \lambda \mathcal{L}_r \tag{9}$$

where $\epsilon_{\theta,t}$ is the model's predicted noise at timestep $t$ for $i$-th subspace, and $\epsilon$ is the actual noise added, and $\mathcal{L}_r = \parallel \sum_{i=1}^{L} \hat{s}_0^{(i)} - x_f \parallel_2^2$, where $\hat{s}_0^{(i)} = s_t^{(i)} - \frac{1-\alpha_t}{\sqrt{1-\bar{\alpha}}}\hat{\epsilon}_\theta$ and $x_f$ is full-sampled image.

In the sampling process, we find a subspace $s$ that is consistent with the measurement signal $y$,

$$\tilde{s} = \min_s \parallel y - A(\sum_i^L s^{(i)}) \parallel_2^2 \tag{10}$$

where $\tilde{s}$ is the subspace that include $\tilde{s}^{(1)}, ..., \tilde{s}^{(L)}$. While Eq. (10) ensures global fidelity via measurement consistency constraint, applying it directly to decoupled subspaces suffers from a one-to-many mapping ambiguity, often leading to unconstrained drift in individual generation trajectories. Here, we adopt a coarse-to-fine strategy to address this issue. We first establish a preliminary subspace-consistent initialization, as follows:

$$\tilde{s}^{(i)} = \min_s(\parallel b^{(i)} - As^{(i)} \parallel_2^2 + \parallel s^{(i)} - \hat{s}_0^{(i)} \parallel_2^2) \tag{11}$$

where $b^{(i)}$ is the k-space of the $i^{th}$ subspace extracted from the undersampled image, $\hat{s}_0^{(i)}$ is computed from Tweedie's formula (Song et al., 2020). The above step anchors the optimization by initializing each subspace to their corresponding measured signals before applying the global constraint in Eq. (10).

In addition, to mitigate the risk of converging to suboptimal local minima, we use the stochastic resampling strategy from (Song et al., 2024), as expressed in the following formulation:

$$p\big(s_t^{(i)} \mid \tilde{s}_t^{(i)}, \tilde{s}_0^{(i)}(y), y\big) = \mathcal{N}\left( \frac{\sigma_t^2\sqrt{\bar{\alpha}_t}\,\tilde{s}_0^{(i)} + (1-\bar{\alpha}_t)\,\tilde{s}_t^{(i)}}{\sigma_t^2 + (1-\bar{\alpha}_t)}, \quad \frac{\sigma_t^2(1-\bar{\alpha}_t)}{\sigma_t^2 + (1-\bar{\alpha}_t)}I \right) \tag{12}$$

where $\sigma_t^2 = \gamma \cdot \frac{1-\alpha_{t-1}}{\alpha_t} \cdot \frac{1-\alpha_t}{\alpha_{t-1}}$ and $\gamma$ represents hyperparameter to control $\sigma^2$, $s_t^{(i)}$ represent posterior sampling based on $\tilde{s}^{(i)}$ as shown in Algorithm 1. The physics-informed guided diffusion generates the subspace $s^{(i)}$, and the final reconstruction is obtained by summing all subspaces as $\hat{x} = \sum_{i=1}^{L} s^{(i)}$.

## 3. Experiments

**Experimental Setup.** The experiments are conducted using the *fastMRI* dataset (Zbontar et al., 2018). The fastMRI data consist of multi-coil $k$-space measurements for a brain image and a single-coil knee image. We transform multi-coil brain images into coil-combine images following the procedure in (Sriram et al., 2020). Specifically, the coil-combine image $x$ is obtained through the corresponding sensitivity maps $S_i$ as, $x = \sum_{j=1}^{N} S_j^* x_j$, where $S_j$ denotes the coil sensitivity map estimated via the *ESPIRiT* algorithm (Uecker et al., 2014), with an auto-calibration region (ACR) factor of 8%. The brain and knee images are cropped to a spatial resolution of $320 \times 320$, and the complex-valued images are split into real and imaginary components, resulting in an input size of $2 \times 320 \times 320$ for the experiments. All images are normalized to the magnitude of the complex value in [0, 1]; thus, the real and imaginary channels are scaled to [-1, 1]. In the experiments, two acceleration factors are

evaluated ($8\times$ and $16\times$). For all acceleration factors, the auto-calibration region (ACR) fraction was set to 4%. The corresponding sampling masks were generated following the official *fastMRI* sampling protocol (Zbontar et al., 2018). Specifically, a fully sampled central k-space region was first retained as the ACR, and the remaining k-space lines outside the ACR were undersampled using the equispaced or random patterns for the brain and knee images. For each module, we train the model for 100 epochs on the entire brain and knee image training dataset. Testing is conducted on 200 randomly selected samples from the brain and knee validation dataset, with one slice extracted from each subject. All modules are trained and tested on a single NVIDIA A100 GPU.

**Evaluation Metrics.** The reconstruction quality was quantitatively evaluated using three standard metrics: peak signal-to-noise ratio (PSNR) in dB, structural similarity index (SSIM), and mean absolute error (MAE). For all metrics, higher PSNR and SSIM values and lower MAE indicate better reconstruction performance. The reported results were computed on magnitude images for both the brain and the knee datasets. For the brain dataset, the evaluation was performed only within the foreground brain region, whereas for the knee dataset, metrics were computed over the entire image. The final results were averaged over the test dataset.

**Implementation Details.** The training of SSDM follows a two-stage process. In the first stage, we train a convolutional dictionary learning (CDL) module for self-reconstruction, following the work (Yang et al., 2025). The CDL module is implemented using a LISTA algorithm unfolded into 12 blocks. In the forward process of the CDL, the input image is transferred into an 8-channel sparse domain and then reconstructed back to the image domain via a reconstruction module $\mathcal{R}$, as shown in Eq.(3).

$$z^{(j+1)} = \text{soft}_{\rho(j)}\big(z^{(j)} + \mathcal{W}_{dec}x - \mathcal{W}_{dec}\mathcal{W}_{conv}z^{(j)} + \mathcal{Q}_2^{(j)}\phi(\mathcal{Q}_1^{(j)}z^{(j)})\big) \tag{13}$$

where $\mathcal{W}_{\text{dec}}$ and $\mathcal{W}_{\text{conv}}$ are convolution operations. $\mathcal{Q}_2$ and $\mathcal{Q}_1$ are $1 \times 1$ convolution operations in the channel dimension, and $\phi$ is the `imGeLU` function. $\rho(j)$ is a learnable soft-threshold using spatial attention that determines the adaptive threshold for each pixel based on the input signal, enabling CDL to process the image with any sampling pattern adaptively. Then, CDL provides unified sparse representations for both fully sampled and undersampled images with any acceleration factor.

In the second stage, we train an unconditional DDPM on the fully sampled subspaces extracted from the frozen CDL module, and we adopt the standard linear noise scheduler defined in DDPM (Ho et al., 2020). Specifically, these subspaces are arranged along the batch dimension as independent training samples, with their complex-valued components separated into distinct channels. Utilizing a U-Net (Dhariwal and Nichol, 2021) as our backbone, we introduce a cross-batch attention mechanism inspired by (Luo et al., 2025) to bridge independent subspaces. To balance performance and computational cost, this attention module is strategically placed only in the middle block and the final encoder block. During inference, the undersampled images are first decomposed into sparse subspaces using the CDL module. These subspaces of undersampled image representations encode physics-related constraints and guide the pretrained diffusion model to generate the corresponding subspaces of the reconstructed image. The inference process is detailed in Algorithm 1. In this algorithm, Eq. (11) is solved using conjugate gradient descent, whereas Eq. (10) is

---

**Algorithm 1:** SSDM MRI Reconstruction

---

**Input:** Measurements $y$. Subspaces measurements $[b^{(1)}, ..., b^{(L)}]$. Sampling mask $M$. Forward operator $A(\cdot, M)$. Stochastic Resample Hyperparameter $\gamma$. Score function $\epsilon_\theta(\cdot, t)$. Noise schedule $\alpha_t$. DDIM parameter $\eta$. Perform resample frequency $C$.

**Output:** $x_0$, reconstructed image

$s_T^{(i)} \sim \mathcal{N}(0, I), \quad i = 1, ..., L$          $\triangleright$ Initial subspaces

**for** $t \leftarrow T - 1$ **to** $0$ **do**

    **for** $i \leftarrow 1$ **to** $L$ **in parallel do**

        $\hat{\epsilon}_t^{(i)} = \epsilon_\theta(s_t^{(i)}, t)$          $\triangleright$ Prediction the score at time-step $t$

        $\hat{s}_0^{(i)} = \dfrac{s_t^{(i)} - \sqrt{1 - \alpha_t}\, \hat{\epsilon}_t^{(i)}}{\sqrt{\alpha_t}}$      $\triangleright$ Computing $\hat{s}_0$ using Tweedie's formula

        $\tilde{s}_0^{(i)} = \min_s(\| b^{(i)} - As^{(i)} \|_2^2 + \| s^{(i)} - \hat{s}_0^{(i)} \|_2^2)$    $\triangleright$ Data consistent in Subspaces

        $\tilde{s}_t^{(i)} = \sqrt{\bar{\alpha}_t}\, \tilde{s}_0^{(i)} + \sqrt{1 - \bar{\alpha}_t}\left(\eta\epsilon_t + \sqrt{1 - \eta^2}\, \hat{\epsilon}_t\right)$    $\triangleright$ Unconditional DDIM step

    **end**

    **if** $t \in C$ **then**

        **for** $i \leftarrow 1$ **to** $L$ **in parallel do**

            $\tilde{s}_0^{(i)}(y) = \min_s \|y - A(\sum_i^L (\tilde{s}_0^{(i)}), M)\|_2^2$    $\triangleright$ Solve $\tilde{s}_0$ in image domain

            $s_t^{(i)} = \text{StochasticResample}(\tilde{s}_0^{(i)}(y), \tilde{s}_t^{(i)}, \gamma)$    $\triangleright$ Map back to $t$

        **end**

    **end**

    **else**

        $s_t = \hat{s}_t(b)$          $\triangleright$ No Resampling

    **end**

**end**

$\hat{x}_0 \leftarrow \sum_i^L (s_0^{(i)})$          $\triangleright$ Output reconstruction

---

solved via standard gradient descent, with its gradients automatically computed through PyTorch's autograd mechanism.

    **Comparing Baseline Methods.** We compared the proposed method with four state-of-the-art diffusion-based methods, including the decomposed diffusion sampler (DDS) (Chung et al., 2024), diffusion posterior sampling (DPS) (Chung et al., 2022), denoising diffusion null-space model (DDNM) (Wang et al., 2022), and Score-Based diffusion models (Score-MRI) (Chung and Ye, 2022). We further compared with traditional compressive sensing with total variation (TV) constraints.

## 4. Results

The results show both the baseline and our proposed reconstructions for acceleration factors of 8×, and 16×. Representative examples of brain and knee images are given in Figure 2, 3, 4. Quantitative evaluations were conducted using SSIM, PSNR, and MAE metrics, as shown in Table 1, 2. Qualitative comparison results for brain MRI at 8× and 16× acceleration factors

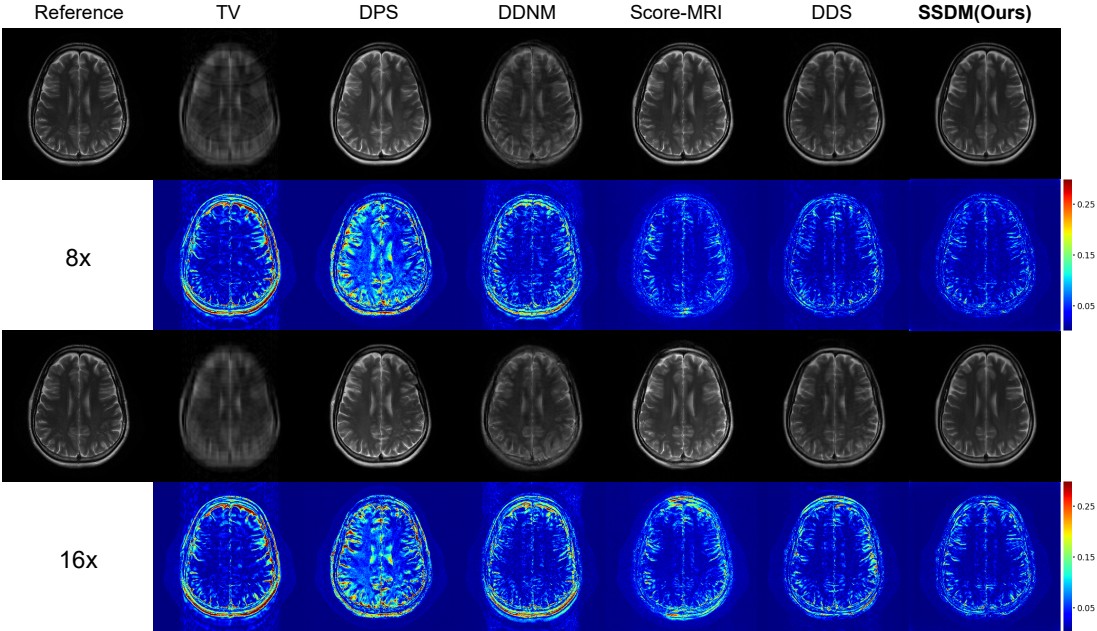

Figure 2: Testing T2-weighted (T2W) brain reconstructions with corresponding error maps at acceleration factors of 8× and 16×. The proposed method demonstrates improved structural preservation and reduced aliasing artifacts compared with baseline methods. All error maps are scaled to [0, 0.3] for display.

are shown in Fig. 2 and Fig. 3. At 8× acceleration, baseline diffusion reconstruction methods can generate visually reasonable results, but tend to attenuate fine gyral patterns and blur tissue interfaces in Fig. 2. Under more aggressive 16× undersampling conditions, these limitations become evident: DDS and Score-MRI exhibit increased smoothing of cortical structures and distinct structural artifacts. As reflected in the corresponding error maps in Fig. 2, errors in these baseline methods are highly correlated with anatomical boundaries. In contrast, SSDM better preserves cortical continuity and fine-grained tissue contrast in both T2- and T1-weighted brain images.

Results for the T2-weighted knee experiment are shown in Fig. 4. Conventional reconstruction methods and DPS exhibit pronounced over-smoothing and residual aliasing artifacts at both 8× and 16× acceleration, leading to blurred cartilage boundaries and loss of fine anatomical details in Fig. 4. While other diffusion-based methods (DDNM, Score-MRI, and DDS) can effectively suppress aliasing, they tend to introduce structural flattening or texture over-smoothing, which becomes particularly apparent at 16× acceleration along cartilage surfaces and joint contours. By comparison, SSDM maintains sharp cartilage boundaries and joint morphology across both acceleration settings, achieving closer visual correspondence with the reference images. The accompanying error maps further demonstrate that SSDM produces minimal residual errors. Unlike baseline methods, which

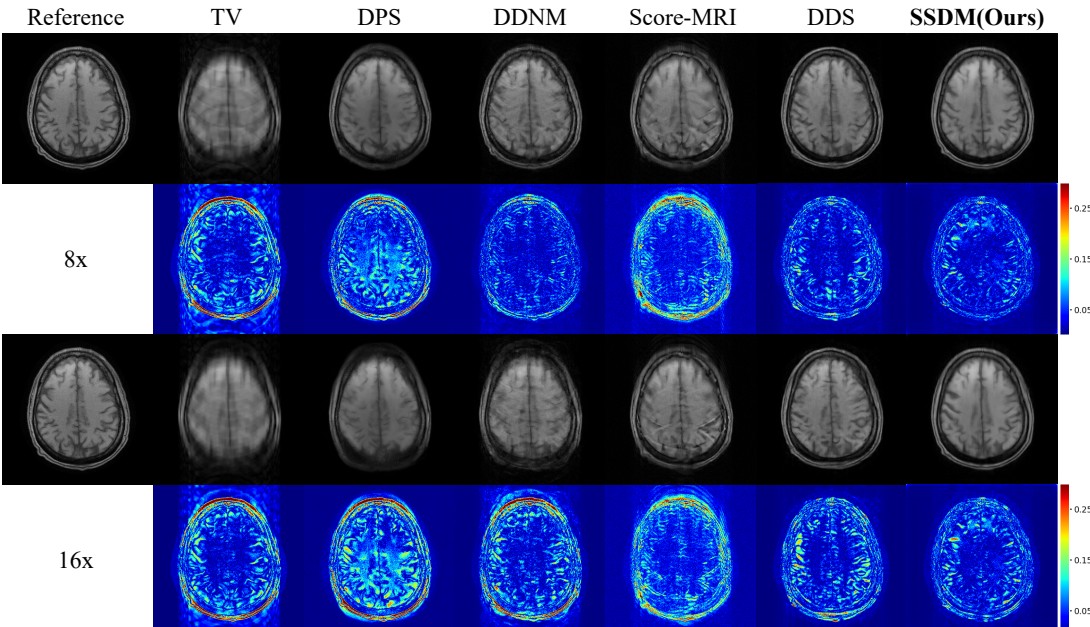

Figure 3: Testing T1-weighted (T1W) brain reconstructions with corresponding error maps at acceleration factors of 8× and 16×. The proposed method demonstrates improved structural preservation and reduced aliasing artifacts compared with baseline methods. All error maps are scaled to [0, 0.3] for display.

Table 1: Quantitative comparison of methods for brain images (mean±std) at 8× and 16× acceleration.

| Method | PSNR (dB) | | SSIM | | MAE ($\times 10^{-3}$) | |
|---|---|---|---|---|---|---|
| | 8× | 16× | 8× | 16× | 8× | 16× |
| TV | 20.18 ± 2.9 | 18.30 ± 2.6 | 0.757 ± 0.064 | 0.734 ± 0.066 | 57.0 ± 18.2 | 55.6 ± 19.3 |
| DPS | 26.67 ± 1.6 | 21.37 ± 1.5 | 0.805 ± 0.044 | 0.798 ± 0.041 | 54.3 ± 16.4 | 55.1 ± 16.2 |
| DDNM | 31.41 ± 1.8 | 23.18 ± 1.8 | 0.836 ± 0.039 | 0.791 ± 0.038 | 49.2 ± 6.2 | 54.1 ± 14.2 |
| Score-MRI | 32.08 ± 2.6 | 27.78 ± 2.7 | 0.813 ± 0.052 | 0.773 ± 0.049 | 54.5 ± 19.7 | 58.9 ± 18.7 |
| DDS | 33.65 ± 2.4 | 31.61 ± 2.3 | 0.923 ± 0.025 | 0.900 ± 0.030 | 41.2 ± 10.9 | 49.7 ± 13.9 |
| **SSDM** | **33.85 ± 2.5** | **32.02 ± 2.6** | **0.927 ± 0.025** | **0.905 ± 0.033** | **40.5 ± 10.7** | **48.1 ± 12.9** |

exhibit prominent structure-correlated errors outlining the anatomy, SSDM's residuals are significantly lower and lack distinct structural patterns.

## 5. Discussion and Conclusion

This work introduces a two-stage diffusion-based MRI reconstruction framework that operates in a learnable sparse space. By integrating coupled convolutional dictionary learning with generative diffusion modeling, the proposed method enables subspace-consistent

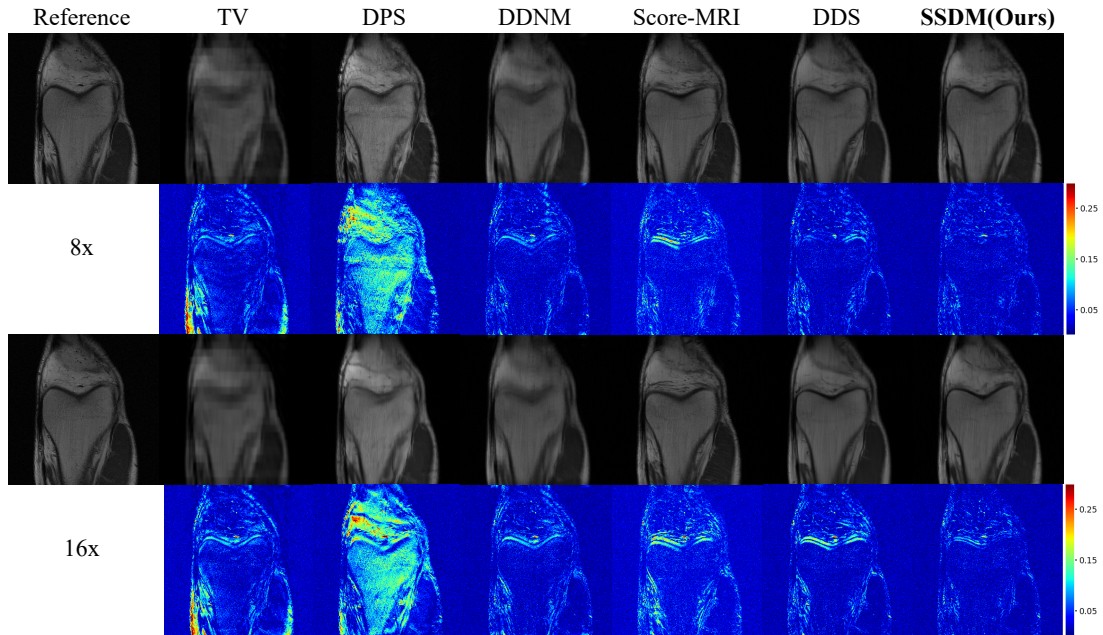

Figure 4: Testing T2-weighted (T2W) knee image reconstructions with corresponding error maps at acceleration factors of 8× and 16×. The proposed method demonstrates improved structural preservation and reduced aliasing artifacts compared with baseline methods. All error maps are scaled to [0, 0.3] for display.

Table 2: Quantitative comparison of methods for knee images (mean±std) at 8× and 16× acceleration.

| Method | PSNR (dB) | | SSIM | | MAE ($\times 10^{-3}$) | |
|---|---|---|---|---|---|---|
| | 8× | 16× | 8× | 16× | 8× | 16× |
| TV | 21.50 ± 3.8 | 21.30 ± 4.7 | 0.680 ± 0.235 | 0.671 ± 0.248 | 40.5 ± 26.6 | 41.5 ± 27.4 |
| DPS | 24.51 ± 2.7 | 21.89 ± 3.9 | 0.779 ± 0.138 | 0.723 ± 0.169 | 30.1 ± 25.9 | 28.2 ± 25.3 |
| DDNM | 29.35 ± 3.6 | 27.97 ± 3.5 | 0.787 ± 0.135 | 0.775 ± 0.139 | 26.0 ± 14.0 | 26.5 ± 11.5 |
| Score-MRI | 29.12 ± 3.1 | 28.37 ± 3.1 | 0.786 ± 0.134 | 0.766 ± 0.136 | 28.7 ± 14.6 | 27.4 ± 12.3 |
| DDS | 29.45 ± 3.4 | 29.13 ± 3.6 | 0.795 ± 0.148 | 0.780 ± 0.146 | 25.2 ± 15.6 | 26.2 ± 14.1 |
| **SSDM** | **29.69 ± 3.2** | **29.21 ± 3.5** | **0.804 ± 0.129** | **0.785 ± 0.134** | **24.6 ± 12.4** | **26.0 ± 13.8** |

reconstruction under measurement constraints, thereby improving fine-detail preservation and suppressing reconstruction artifacts.

Aliasing artifacts in accelerated MRI arise from energy leakage of underlying signals, leading to severe entanglement of structural information under highly undersampled conditions and posing a fundamental challenge for both discriminative and generative reconstruction models. By decomposing the complex image representation into multiple sparse subspaces, our framework promotes effective disentanglement of signal components. It supports faithful subspace-wise reconstruction, mitigating the risk of generating perceptually

plausible but feature-inconsistent results. We note that this framework targets a complementary regime to fully supervised regression or unrolled reconstruction methods, focusing on unconditional generative priors under extreme undersampling rather than paired input–output supervision.

However, the current implementation is limited to single-coil 2D Cartesian acquisitions. Future work will extend the framework to multi-coil and dynamic MRI settings and further investigate richer learnable physics-informed spaces to incorporate more expressive physical priors. Overall, this study demonstrates that coupling diffusion priors with structured sparse subspace representations provides a practical pathway for leveraging physical signal properties in the transform space, thereby enhancing MRI reconstruction quality.

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

## Appendix A. Ablation Study

### A.1. Analysis of Stage 1: Convolutional Dictionary Learning

We investigated the trade-offs among sparse representation capacity, sparse features as shown in 5 and 6, computational efficiency, and feature expressiveness by analyzing the sparse channel configuration and the thresholding strategy.

**1. Sparse Channel Capacity and Dead Channels**  We evaluated the impact of the sparse channel dimension ($N_c$) by testing configurations with $N_c \in \{4, 8, 16\}$.

- **Low Channel Count ($N_c = 4$):** We observed that reducing the number of channels restricts the model's ability to decompose complex signals into sufficiently simplified components. The limited capacity forces the model to encode dense information into fewer channels, resulting in a complex representation in sparse features.

- **High Channel Count ($N_c = 16$):** While increasing the number of channels theoretically enhances representational capacity, we observed the **'dead channel'** phenomenon, in which a subset of filters fails to capture meaningful signals, leading to persistent inactivity. Furthermore, a larger $N_c$ significantly increases the computational burden during the second-stage diffusion process without yielding proportional performance gains.

- **Optimal Setting ($N_c = 8$):** We found that $N_c = 8$ strikes an optimal balance, maintaining excellent sparsity properties and representation quality while avoiding dead channels and high computational costs.

**2. Sparsity-Fidelity Trade-off via Soft-Thresholding**  We analyzed the effect of the initial threshold value in the learnable soft-thresholding function on the sparsity rate.

- **Increasing the Threshold:** By raising the initial threshold for the learnable soft-threshold function, we forced more pixel signals to be suppressed to zero, effectively increasing the sparsity rate of the latent representation.

- **Impact on Reconstruction:** However, we found that the image representation relies heavily on the expressiveness of these sparse features. Excessive suppression (via a high threshold) limits this expressiveness, degrading the quality of self-reconstruction. Therefore, the threshold is initialized at 1 and scales down by a factor of 0.8 for each subsequent block.

### A.2. Analysis of Stage 2: Subspace Diffusion Model

In the second stage, we validated the architectural design with respect to global consistency and the impact of the sampling trajectory.

**1. Cross-Batch Attention and Manifold Consistency**  We investigated the critical role of the cross-batch attention mechanism in coordinating the generation of multiple subspaces.

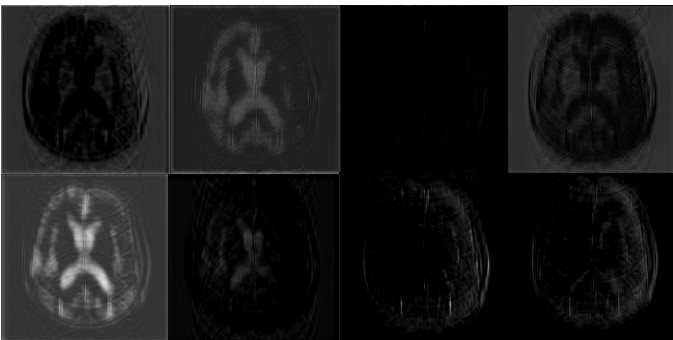

Figure 5: The sparse features of the under-sampled image at 8× acceleration.

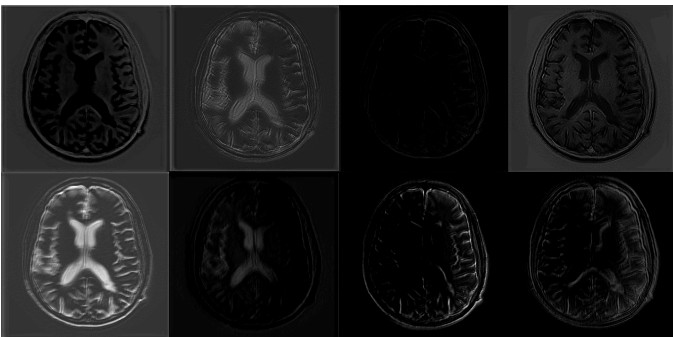

Figure 6: The sparse features of the fully sampled reference image.

Since the subspaces are decomposed parts of a single anatomical structure, their generation trajectories must be coupled to ensure they converge onto a unified image manifold. When the cross-batch attention was removed, the subspaces were generated independently without global guidance. This lack of coupling led to spatial misalignment between subspaces. The resulting aggregation of misaligned subspaces led to severe blurring and loss of coherence, confirming the necessity of an information-exchange pathway to ensure global spatial consistency.

### A.3. Performance Statistics

Table 3 summarizes the model parameter count, the single-forward runtime of the backbone network, and the end-to-end GPU runtime per slice for different reconstruction methods. Due to their intrinsic inference mechanisms, the compared methods employ different numbers of sampling steps. Specifically, DPS and Score-MRI follow standard diffusion sampling with 1000 steps, whereas DDNM, DDS, and SSDM involve deterministic sampling strategies that require only 100 steps. All methods are evaluated under their native inference settings rather than enforcing a unified step count. Since DPS and DDNM use the same diffusion backbone as DDS, we do not separately report their model parameter size and NFE and instead provide only their end-to-end runtimes for comparison. All experiments are conducted on a single NVIDIA A100 GPU.

Table 3: Model complexity and inference efficiency comparison.

| Model | Parameters (M) | ms/NFE | Sec/Slice |
|-------|----------------|--------|-----------|
| DPS | - | - | 199.68 |
| DDNM | - | - | 12.0 |
| Score-MRI | 61.4 | $117.66 \pm 0.54$ | 498.05 |
| DDS | 361.4 | $95.82 \pm 0.19$ | 13.30 |
| SSDM(Ours) | 367.4 | $281.57 \pm 0.28$ | 86.02 |

Although Score-MRI employs a lightweight diffusion backbone, its end-to-end runtime remains high (498.05 s/slice) due to the large number of sampling steps. In contrast, DDS substantially reduces the total number of network function evaluations (NFE) and achieves a significant speedup (13.30 s/slice) despite using a larger backbone. SSDM further extends DDS by introducing subspace-wise multi-stage sampling, which increases both the per-forward computational cost and the total NFE.

Overall, these results indicate that SSDM trades increased computational cost for improved reconstruction performance. It decomposes a complex reconstruction problem into multiple simpler subproblems solved in parallel, resulting in a runtime within the same order of magnitude as DDS but higher, while maintaining comparable model capacity.

