# OpenReview forum: "Sparse Subspace Diffusion Model for Physically Consistent Accelerated MRI Reconstruction"
_MIDL.io/2026/Conference — MIDL 2026 Poster_

### Official Review · Reviewer_hrht · 2026-01-01

**Confidence:** 3
**Preliminary Rating:** 3
**Final Rating:** 4

**Summary:**

The authors introduce the Sparse Subspace Diffusion Model (SSDM), a two-stage framework for accelerated MRI reconstruction that operates in an adaptive sparse space. In the first stage, convolutional dictionary learning is used to decompose MR images into multiple sparse and approximately orthogonal subspaces, yielding a unified multi-channel sparse representation. In the second stage, an unconditional diffusion model is applied to each sparse subspace, with cross-subspace interaction and physics-informed guidance enforcing measurement consistency during sampling. Experiments on the fastMRI brain and knee datasets at 8× and 16× acceleration demonstrate improved PSNR, SSIM, and preservation of fine anatomical structures compared to existing diffusion-based and compressed-sensing baselines.

**Strengths:**

1. The idea of reconstructing images from individual sparse subspaces is interesting and well motivated.
2. The physics-informed constraints, including the orthogonality constraint $L_{\perp}$ and the coupling constraint $L_C$, are potentially helpful.
3. The proposed method shows consistently improved visual quality, with better preservation of fine anatomical details and reduced aliasing artifacts compared to baseline methods.

**Weaknesses:**

1. The computational cost may be higher than standard diffusion-based reconstruction methods, since the framework requires multiple transformations, subspace decomposition, and diffusion sampling across several sparse subspaces.
2. Lack of ablation study on the effectiveness of $L_{\perp}$ and $L_C$. Moreover, the impact of the number of sparse channels is only discussed without quantitative experimental support.
3. In addition, I'm curious to see if the method can be better if stage 1 and 2 can be trained end-to-end instead of sequentially.

**Detailed Comments:**

The term physics-informed remains somewhat vague, as the physical modeling mainly relies on enforcing orthogonality between sparse subspaces and k-space data consistency, rather than explicitly embedding MRI physics into the diffusion model itself.

**Justification Of Final Rating:**

While a more detailed ablation showing the contribution of individual components would further be helpful to the paper, I understand the limited time constraint. The increased computation time from DDS to the proposed method is less than I expected, and the explanation regarding the infeasibility of fully end-to-end training is reasonable. I have raised my score to weak accept.

**Justification Of The Preliminary Rating:**

The paper presents an interesting and well-motivated idea of performing diffusion-based MRI reconstruction in learned sparse subspaces and demonstrates strong qualitative and quantitative results on fastMRI. However, the lack of ablation studies on the effectiveness of the physics-informed constraints makes it difficult to assess the individual contributions of these components. In addition, the proposed method is potentially more computationally expensive, as it requires multiple diffusion models, while the quantitative improvements over competing methods are relatively modest.

**Questions To Address In The Rebuttal:**

1. Report the inference time of the proposed method vs. competing methods.
2. Add ablation studies on removing (1) $L_{\perp}$ only (2) $L_C$ only (3) $L_{\perp} + L_C$. Report the change on all metrics (MAE, SSIM, PSNR).
3. Consider training Stage 1 (CDL decomposition) and Stage 2 (subspace diffusion) jointly in an end-to-end manner if feasible.

---

> ### Author Response · Authors · 2026-01-24
> **We thank the Reviewer hrht for the invaluable comments. We start the rebuttal by answering these comments and following up with response R to specific questions Q.**
>
> Q1-R1: We will include additional statistics on computational efficiency in the ablation study. Please see table 3 in ablation study at A.3 Performance Statistics.
>
> Q2-R2: Due to limited time in Rebuttal period, we will add these experiments to the ablation study and update the results in the final manuscripts.
>
> Q3-R3: We thank the reviewer for the suggestion of end-to-end joint training. In practice, however, a two-stage training strategy is widely adopted in latent diffusion-based generative models, where representation learning and diffusion modelling are decoupled to improve training stability, controllability, and computational efficiency.
> In our setting, Stage 1 (CDL-based decomposition) is designed to learn a stable, structured, sparse representation, while Stage 2 focuses on generative modelling within this representation space. Joint training would significantly increase both the difficulty of optimization and the computational cost. We therefore adopt a modular two-stage design, which is both practical and consistent with prior diffusion literature; exploring fully end-to-end joint training under larger computational budgets is an interesting direction for future work.

---

### Official Review · Reviewer_tjqf · 2026-01-07

**Confidence:** 4
**Preliminary Rating:** 5
**Final Rating:** 5

**Summary:**

This paper introduces a new model for MR image reconstruction called the Sparse Subspace Diffusion Model (SSDM). SSDM is a two-stage approach to reconstruct equidistantly or randomly undersampled, accelerated MRI data (all simulations). The core idea is to bridge physics-informed sparse representations with a diffusion-based generative model.
Fristly, a coupled convolutional dictionary learning (CDL) decomposes MR images into multiple orthogonal sparse subspaces. Secondly, an unconditional diffusion model operates within these subspaces, guided by measurement-consistency constraints during inference.
Data used originates from the publicly available fastMRI dataset.

**Strengths:**

Overall, the abstract is well written and suggest an interesting and party novel approach, i.e. the suggested architecture to promote disentanglement of different signal components in latent respresentation. The methods are well described and necessary mathematical background is detailed. The experiments appear complete (based on data available).

**Weaknesses:**

The results presented in common metrics for medical imaging (ssim, psnr, mae) show only modest superiority of SSDM compared to other models. Ideally, test & validation data would have been kept totally separated from training data (different subjects). The data-split strategy is not clearly stated in the abstract.

Model performance was not evaluated on "real data" - no own measurements or truly undersampled MR data.

**Detailed Comments:**

Thank you for submitting your work to MIDL. I would like to share some thoughts:
- Please provide some more details on the data used: Did you pay attention to the subjects for data splitting (train, val, test) - multiple slides belong to a single subject?
- Assuming the evaluation metrics are mean over all test data -> please provide SD.
- Why did you mask brain (only foreground) and knee (entire image) images differently for evaluation?
- Please provide information about the consistency of the performance? E.g. is SSDM performing similar in superior/inferior slices in the brain?

**Justification Of Final Rating:**

Thanks to the authors for addressing the reviewers comments. I will leave my rating unchanged. I can see some points of the other reviewers. Overall the manuscript was convincing to me and worth to be considered at MIDL.

**Justification Of The Preliminary Rating:**

The paper presents solid work with sufficient novelty. The methods are presented well and model performance matches/slightly surpasses other approaches. The model architecture is relevant to the field and is worth presenting to the community and a broad scientific discussion.

**Questions To Address In The Rebuttal:**

/

---

> ### Author Response · Authors · 2026-01-24
> **We thank the Reviewer tjqf for the invaluable comments. We start the rebuttal by answering these comments and following up with response R to specific questions Q.**
>
> Q1-R1:We will include the dataset split in the paper.  We used the official fastMRI dataset split for training and validation. Please see the details in Experimental Setup in Experiments.
> During training, we used only the fastMRI training set without mixing in any other data. For testing, we used the fastMRI validation dataset, from which we randomly sampled 200 test cases and randomly selected individual slices. So multiple slice belong to different subjects. No subject overlap exists between training and test sets.
>
> Q2-R2: We will update the results with the test data's standard deviation.
>
> Q3-R3:In brain images, the background occupies most of the area in the image, so computing evaluation metrics based on the entire image would lead to inflated scores; therefore, only the foreground region is computed. In knee images, the foreground occupies the main part of the image, so the entire image is used for computation.
>
> Q4-R4: Slices are randomly sampled across the entire volume, covering both superior and inferior regions, without manual selection. In our experiments, we did not observe systematic degradation in superior or inferior slices, and the qualitative improvements of SSDM are consistently present across different anatomical locations. We add more detail in the Experiment Setup in the Experiment, and highlighted in red.

---

### Official Review · Reviewer_AuNL · 2026-01-09

**Confidence:** 4
**Preliminary Rating:** 3
**Final Rating:** 4

**Summary:**

This paper proposes Sparse Subspace Diffusion Model (SSDM) for accelerated MRI reconstruction. The method uses a two-stage pipeline: (i) a coupled convolutional dictionary learning (CDL) module implemented via unfolded LISTA decomposes an image into L=8 sparse/orthogonal subspaces; (ii) an unconditional diffusion model is trained on fully sampled subspaces, with a cross-batch attention mechanism to exchange information across subspaces. During inference, the reverse diffusion is guided by measurement consistency using a coarse-to-fine strategy that enforces subspace-level and global data consistency. Experiments are conducted on fastMRI brain and knee at 8× and 16× accelerations.

**Strengths:**

Interesting hybridization of sparse coding and diffusion priors. Using an adaptive sparse transform (CDL/LISTA) to define the diffusion space is a reasonable attempt to reduce entanglement in high-acceleration settings.

Consistent improvements over diffusion-based baselines on both brain and knee, especially at 16×.

The paper includes some ablation/analysis on sparse channel capacity and the necessity of cross-subspace coupling via attention.

**Weaknesses:**

The “physically consistent / physics-informed” claim is overstated due to the single-coil proxy setup Although the method enforces data consistency under $A=M F$, the experiments operate on coil-combined complex images, where multi-coil physics and parallel imaging encoding are absent.

Missing strong MRI reconstruction baselines (regression/unrolled SOTA), making competitiveness unclear. Accelerated MRI reconstruction has well-established high-performing regression/unrolled baselines (e.g., VarNet variants, modern transformer-based reconstructions)[1-3].

The core concept “diffusion in orthogonal subspaces for MRI reconstruction” appears related to prior work such as Sub-DM[4]. More generally, “subspace diffusion” has been studied in the generative modeling literature[5].

The subspace decomposition may mainly act as increased capacity/ensemble; key capacity-matched ablations are missing
The method effectively generates and aggregates L=8 subspace images. While Appendix A analyzes channel counts and attention removal, there is no decisive ablation against:

- a single larger diffusion model with matched parameter count / FLOPs,
- a naive ensemble of diffusion models,
- fixed-transform subspaces (wavelet bands) vs learned CDL subspaces,

The quantitative gains over DDS are relatively small, yet SSDM involves a two-stage training pipeline and iterative optimization within sampling. Could the authors report the runtime and more performance detail.


SSDM relies on a learned dictionary-based sparse subspace decomposition, which is less directly tied to the MRI forward model and may be less robust under domain shifts or unseen masks. In contrast, PDAC [6] uses a more physics-aligned degradation/subsampling decomposition with progressive reconstruction; a brief discussion and a cross-mask robustness comparison would strengthen the justification for SSDM’s design choice.


[1] Guo, P., Mei, Y., Zhou, J., Jiang, S., & Patel, V. M. (2023). Reconformer: Accelerated mri reconstruction using recurrent transformer. IEEE transactions on medical imaging, 43(1), 582-593.
[2] Sriram, A., Zbontar, J., Murrell, T., Defazio, A., Zitnick, C. L., Yakubova, N., ... & Johnson, P. (2020, September). End-to-end variational networks for accelerated MRI reconstruction. In International conference on medical image computing and computer-assisted intervention (pp. 64-73). Cham: Springer International Publishing.
[3] Fabian, Z., Tinaz, B., & Soltanolkotabi, M. (2022). Humus-net: Hybrid unrolled multi-scale network architecture for accelerated mri reconstruction. Advances in Neural Information Processing Systems, 35, 25306-25319.
[4] Guan, Y., Cai, Q., Li, W., Fan, Q., Liang, D., & Liu, Q. (2024). Sub-DM: Subspace Diffusion Model with Orthogonal Decomposition for MRI Reconstruction. arXiv preprint arXiv:2411.03758.
[5] Jing, B., Corso, G., Berlinghieri, R., & Jaakkola, T. (2022, October). Subspace diffusion generative models. In European conference on computer vision (pp. 274-289). Cham: Springer Nature Switzerland.
[6] Wang, C., Guo, L., Wang, Y., Cheng, H., Yu, Y., & Wen, B. (2024). Progressive divide-and-conquer via subsampling decomposition for accelerated mri. In Proceedings of the IEEE/CVF Conference on Computer Vision and Pattern Recognition (pp. 25128-25137).

**Detailed Comments:**

Please see weakness

**Justification Of Final Rating:**

Please refer to Comments to Authors.

The authors present a mathematically grounded, non-trivial decomposition strategy. Although the limitations regarding coil-combined inputs and the omission of supervised baselines is still hold, which are currently common issues in the generative MRI sub-field, rather than flaws unique to this work.

**Justification Of The Preliminary Rating:**

Why not higher? The CDL-based subspace diffusion idea is well-motivated and improves over several diffusion baselines, but a few gaps limit confidence in the overall claim. The experiments rely on ESPIRiT coil-combined, simulated single-coil data, so the “physically consistent” scope would benefit from clearer framing relative to standard multi-coil PI. Comparisons are mostly within diffusion methods; adding a strong regression/unrolled baseline (e.g., VarNet-class) would better contextualize performance. Finally, capacity-matched ablations (single larger model/ensemble) and basic runtime + cross-mask robustness reporting would help justify the subspace design and practicality.

**Questions To Address In The Rebuttal:**

If train a single diffusion model with matched parameters (no CDL decomposition) or an 8-model ensemble, will still observe SSDM’s gains?

---

> ### Comment · Reviewer_AuNL · 2026-01-24
> **Comments to Authors**
>
> I thank the authors for their detailed response and clarifications. The rebuttal has addressed several of my technical concerns, particularly regarding the model capacity and the nature of the subspace decomposition.
>
> - The clarification that weights are shared is crucial. It confirms that performance gains stem from the effectiveness of the sparse subspace decomposition strategy itself, rather than naive parameter scaling or ensembling.
>
> However, I retain the following reservations:
>
> - Clinical utility depends on quality and speed, regardless of the training paradigm. Being "unconditional" does not exempt the method from comparison with SOTA supervised baselines. Similar generative works like [1] explicitly compare against supervised methods to contextualize performance.
>
> - I maintain that enforcing consistency on ESPIRiT-combined data is physically sub-optimal, as it alters noise distributions and discards parallel imaging encoding. However, I acknowledge this is currently a common practice in generative MRI research.
>
> The authors present a mathematically grounded, non-trivial decomposition strategy. I recognize that the limitations regarding coil-combined inputs and the omission of supervised baselines are currently common issues in the generative MRI sub-field, rather than flaws unique to this work. As the shared-weight design addresses my primary technical concern regarding model redundancy, I will raise my score to 4.
>
> [1] Chung, H., & Ye, J. C. (2022). Score-based diffusion models for accelerated MRI. Medical Image Analysis, 80, 102479.

---

### Author Response · Authors · 2026-01-24
**We thank the Reviewer AuNL for the invaluable comments. We start the rebuttal by answering these comments and following up with response R to specific questions Q from each Rreviewer.**

Q1-R1: In this work, ‘physical consistency’ refers specifically to data consistency in the MRI forward model. We have clarified this further by rephrasing 'physically consistent' to 'measurement-consistent' and highlighted it in red. We acknowledge that our experiment uses 'coil combined' data; our fundamental approach can be readily extended to multi-coil settings, and it embeds ESPRiT, a SOTA parallel imaging method.

Q2-R2:We acknowledge that fully supervised regression/unrolled methods such as VarNet or Reconformer typically achieve strong performance when paired supervision is available, e.g., paired under-sampled input and fully sampled output. However, this work focuses on a different scope and our approach employs unconditional generative priors, i.e., we do not use under-sampled input or as conditions, so this approach aims to work with varying under-sampling factors. To address the reviewer’s comment, we have added further discussion to clarify this aspect.

Q3-R3:We agree that subspace decomposition is also adopted in Sub-DM [4], prior subspace diffusion models [5], and our work; however, the definition of subspaces and their role in diffusion fundamentally differ. Sub-DM [4] employs wavelet decomposition to separate frequency components, treating them as channels and jointly modelling them within a single diffusion trajectory. This decomposition mainly reshapes the data geometry to stabilise the score field, rather than simplifying the learning instances themselves. In contrast, subspace diffusion models [5] define subspaces as noise-scale–dependent nested supports, where modelling is restricted to low-dimensional subspaces at high noise levels and progressively expanded during sampling.

Unlike these approaches, we redefine subspaces as independent generative learning instances rather than as feature channels or noise-dependent nested subspaces. During training, each image is decomposed into multiple sparse subspaces, treated as independent samples, and organised along the batch dimension. A single shared diffusion backbone models these subspaces in parallel, while cross-batch attention explicitly couples their evolution to preserve global structure. Under extreme undersampling, this instance-level subspace formulation transforms the generation of a complex image distribution into the parallel generation of simpler instances, which are then fused to obtain the final reconstruction. We have added clarifying text to the Introduction and highlighted it in red.

Q4-R4:Although SSDM operates on multiple sparse subspaces, all subspaces are processed by a single shared diffusion backbone with shared parameters. The goal is not to increase model capacity through independent models, but to restructure the learning problem by decomposing a complex image distribution into simpler, sparse components. Compared to the original DDPM backbone, we introduce only two lightweight cross-batch attention modules in the encoder's last and middle blocks to facilitate information exchange across different subspaces, with a spatial complexity of $O\left(B\cdot{H}\cdot{W}\cdot{S^2}\right)$. The diffusion module's capacity remains the same. Please see the details in the ablation study at A.3 Performance Statistics.

Q5-R5:We will include additional statistics on computational efficiency in the ablation study. Please see Table 3 in the ablation study at A.3 Performance Statistics.

Q6-R6:We agree that the sparse subspaces learned by CDL are not directly derived from the MRI forward model. Our goal is not to align the subspace decomposition with the forward operator, but to construct a representation that is more favourable for generative modelling under extreme undersampling; physical consistency is enforced during inference via data consistency with the forward model.

SSDM does not rely on a fixed sampling mask. Random masks are used during both training and inference to reduce overfitting. In the first-stage self-reconstruction, conventional soft-thresholding is replaced by a pixel-wise adaptive spatial attention mechanism conditioned on the input, allowing flexible adaptation to different sampling patterns. Although this does not provide a formal robustness guarantee under all domain shifts, empirical results show stable performance across diverse random masks.

We also acknowledge that PDAC [6] adopts a more physics-aligned progressive reconstruction strategy. In contrast to sequentially solving nested degradation subproblems, our method decomposes the image into multiple simplified and independent learning instances that are modelled in parallel by a shared diffusion backbone, reducing the complexity of individual generative tasks without embedding the forward model into the subspace definition.

Questions To Address In The Rebuttal:
Please see the response for Q4

---

### Author Rebuttal · Authors · 2026-01-24

**Rebuttal:**

Please see the details in the response to each reviewer's comments and the modified manuscript.

**Supporting Material:**

/attachment/84eb763447e1eb372f4082f0e0dfcfd774e764bc.pdf

---

### Meta-Review · Area_Chair_Rz6t · 2026-02-01

**Recommendation:** Accept (Oral)
**Confidence:** 4

**Metareview:**

I agree with the merits mentioned by the reviewers. The work can be accepted as an oral presentation, given that the rebuttal has been done quite well. (One review comment has been deleted by the PC, not sure why and I didn't consider the review's comment)

---

### Decision · Program_Chairs · 2026-02-13

Accept (Poster)